# TOWARDS INTERPRETABLE MOLECULAR GRAPH REPRESENTATION LEARNING

## ABSTRACT

Recent work in graph neural networks (GNNs) has led to improvements in molecular activity and property prediction tasks. Unfortunately, GNNs often fail to capture the relative importance of interactions between molecular substructures, in part due to the absence of efficient intermediate pooling steps. To address these issues, we propose LaPool (Laplacian Pooling), a novel, data-driven, and interpretable hierarchical graph pooling method that takes into account both node features and graph structure to improve molecular understanding. We benchmark LaPool and show that it not only outperforms recent GNNs on molecular graph understanding and prediction tasks but also remains highly competitive on other graph types. We then demonstrate the improved interpretability achieved with LaPool using both qualitative and quantitative assessments, highlighting its potential applications in drug discovery.

## 1 INTRODUCTION

Following the recent rise of deep learning for image and speech processing, there has been great interest in generalizing convolutional neural networks to arbitrary graph-structured data (Gilmer et al., 2017; Henaff et al., 2015; Xu et al., 2018). To this end, graph neural networks (GNN) falling into either spectral-based or spatial-based approaches have been proposed. Spectral methods define the graph convolution (GC) as a filtering operator of the graph signal (Defferrard et al., 2016), while spatial methods define the GC as a message passing and aggregation across nodes (Henaff et al., 2015; Xu et al., 2018; Jin et al., 2018). In drug discovery, GNNs have been very successful across several molecular graph classification and generation tasks. In particular, they outperform predetermined molecular fingerprints and string-based approaches for molecular property prediction and the *de novo* generation of drug-like compounds (Jin et al., 2018; Li et al., 2018b).

However, the node feature update performed by GNNs introduces important limitations. For instance, experimental results indicate a performance decrease for deeper GNNs due to the signal smoothing effect of each GC layer (Li et al., 2018a). This limits the network's depth and restricts the receptive field of the vertices in the graph to a few-hop neighborhood, which is insufficient to properly capture local structures, relationships between nodes, and subgraph importance in sparse graphs such as molecules. For example, at least three consecutive GC layers are needed for atoms at the opposite side of a benzene ring to exchange information. This issue is exacerbated by the single global pooling step performed at the end of most GNNs that ignores any hierarchical structure within the graph.

To cope with these limitations, graph coarsening (pooling) methods have been proposed to reduce graph size and enable long-distance interaction between nodes. The earliest proposed methods relied solely on deterministic clustering of the graphs, making them non-differentiable and task-independent (Jin et al., 2018; Dafna and Guestrin, 2009; von Luxburg, 2007; Ma et al., 2019). In contrast, more recent methods use node features but, as we will show, are unable to preserve the graph structures after pooling (Ying et al., 2018; Gao and Ji, 2018), limiting their interpretability.

Borrowing from theory in graph signal processing, we propose LaPool (Laplacian Pooling), a differentiable pooling method that takes into account both the graph structure and its node features. LaPool performs a dynamic and hierarchical segmentation of graphs by selecting a set of centroid nodes as cluster representatives (centroids) using the graph Laplacian, then learns a sparse assignment of the remaining nodes (followers) into these clusters using an attention mechanism. The primary contributions of this paper are summarized below:

- We propose a novel and differentiable pooling module (LaPool) that can be incorporated into existing GNNs to yield more expressive networks for molecular data.
- We propose a graph structure understanding dataset for benchmarking GNNs that is based on molecular substructure prediction.
- We show that LaPool outperforms recently proposed graph pooling layers on both discriminative and generative molecular graph benchmarks, while also remaining competitive on other graph benchmarks.
- We highlight the improved interpretability achieved by LaPool using both qualitative and quantitative assessments.

We argue that the enhanced performance and interpretability achieved by LaPool can improve our understanding of molecular structure-activity relationships, and therefore has important applications in drug discovery.

## 2 RELATED WORK

In this section, we first introduce related work on graph pooling, then provide an overview of techniques used in computational drug discovery to put our work into context. As our focus herein is on graph pooling, we refer readers to (Wu et al., 2019) for an overview of recent progress in GNNs.

In traditional GNN architectures, global sum/average/max-pooling layers have been used to aggregate node embeddings into a graph-level representation. Recently, more sophisticated methods have been proposed. For example, Li et al. (2015) uses a gated mechanism, Zhang et al. (2018) proposed SortPool which sorts node features before feeding them into a 1D convolution, while in (Gilmer et al., 2017) node feature averaging was substituted by a Set2Set architecture. Although these new global aggregation methods have been shown to outperform standard global pooling, they completely overlook the rich structural information on graphs which has been shown as necessary for building effective GNN models (Ying et al., 2018; Ma et al., 2019).

Consequently, hierarchical graph pooling methods have been proposed. They act by reducing graph size and increasing node receptive fields without increasing network depth. However, in contrast to the regular structure of images, graphs are irregular and complex, making it challenging to properly pool nodes together. Certain hierarchical graph pooling methods therefore rely on deterministic and non-differentiable clustering to segment the graph (Defferrard et al., 2016; Jin et al., 2018). More recently, differentiable hierarchical graph pooling layers have been proposed. Ying et al. (2018) proposed DiffPool, a pooling layer that performs a similarity-based node clustering using a soft affinity matrix learned by a GNN. Likewise, Graph U-Net was proposed in (Gao and Ji, 2018) as a sampling method that retains and propagates only the top-k nodes at each pooling step based on a learned importance value.

In computer-aided drug discovery, methods such as virtual screening and *de novo* drug design serve as efficient complements to physical high-throughput screening of large molecular libraries. For example, virtual screening, which aims to accurately predict molecular properties directly from molecular structure, can play an important role in rapidly triaging promising compounds early in drug discovery (Subramaniam et al., 2008). Importantly, data-driven virtual screening approaches that leverage advances in deep learning, rather than pre-determined features such as molecular fingerprints (Rogers and Hahn, 2010) and SMILES string representations, have been shown to dramatically improve prediction accuracy (Kearnes et al., 2016; Wu et al., 2018). Similarly, advances in generative models have enabled the application of deep generative techniques such as VAE (Kingma and Welling, 2013) and GAN (Goodfellow et al., 2014) to the *de novo* design of drug-like molecules. The first molecular generative models (e.g. Grammar-VAE (Kusner et al., 2017)) resorted to generating string representations of molecules (via SMILES), which resulted in a high proportion of invalid structures due to the complex syntax of SMILES. Graph generative models have since been developed (e.g. JT-VAE (Jin et al., 2018), GraphVAE (Simonovsky and Komodakis), MolGAN (De Cao and Kipf, 2018), MolMP (Li et al., 2018b), etc.) and have been shown to improve the validity and novelty of generated molecules. In addition, these methods allow conditional molecule generation via Bayesian optimization or reinforcement learning (Jin et al., 2018; Olivecrona et al., 2017; Assouel et al., 2018; Li et al., 2018d; You et al., 2018a). In this work, we are mainly interested in the impact of molecular representation on generative performance as opposed to the optimization procedure itself.

# 3 GRAPH LAPLACIAN POOLING

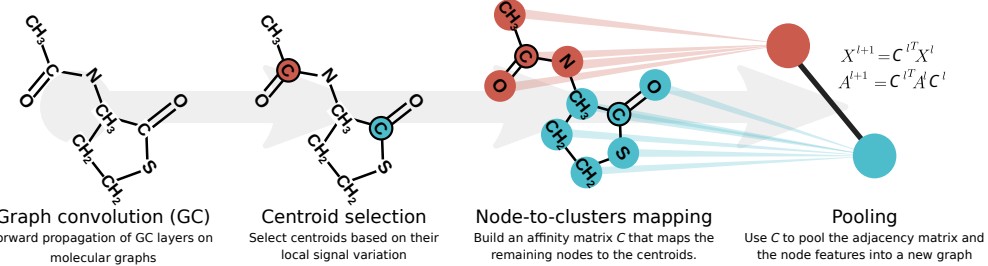

**Figure 1:** Overview of the proposed Laplacian Pooling (LaPool) method

A reliable pooling operator should maintain the overall structure and connectivity of a graph. LaPool achieves this by taking into account the local structure defined by the neighborhood of each node. As shown in Figure 1, the method uses a standard GC layer with a centroid selection and a follower selection step. First, the centroids of the graph are selected based on the local signal variation (see Section 3.2). Next, LaPool learns an affinity matrix $C$ using a distance normalized attention mechanism to assign all nodes of the graph to the centroids (see Section 3.3). Finally, the affinity matrix is used to coarsen the graph. These steps are detailed below.

## 3.1 PRELIMINARIES

**Notation** Let $G = \langle V, A, X \rangle$ be an undirected graph, where $V = \{v_1, \ldots v_n\}$ is its vertex set, $A \in \{0, 1\}^{n \times n}$ denotes its adjacency matrix, and $X = [\mathbf{x_1}, \ldots \mathbf{x_n}]^\mathsf{T} \in \mathbf{R^{n \times d}}$ is the node feature matrix with each node $v_i$ having $d$-dimensional feature $x_i$. $X$ can also be viewed as a $d$-dimensional signal on $G$ (Shuman et al., 2012). Without loss of generality we may assume a fixed ordering of the nodes that is respected in $V$, $A$, and $X$. The neighborhood of radius $h$ (or $h$-hop) neighborhood of a node $v_i \in V$ is the set of nodes separated from $v_i$ by a path of length at most $h$ and is denoted by $\mathcal{N}^h(v_i)$. For simplicity, we will use $\mathcal{N}(v_i)$ to refer to the set of nodes adjacent to $v_i$.

**Graph Signal** For any graph $G$, its graph Laplacian matrix $L$ is defined as $L = D - A$, where $D$ is a diagonal matrix with $D_{i,i}$ being the degree of node $v_i$. The graph Laplacian is a difference operator and can be used to define the smoothness $s(X)$ (the extent at which the signal changes between connected nodes) of a signal $X$ on $G$. For 1-dimensional signal $\mathbf{f} = [f_1, \ldots, f_n]$ :

$$s(\mathbf{f}) = (\mathbf{f}^\mathsf{T} L \mathbf{f}) = \frac{1}{2} \sum_{i,j}^{n} A_{i,j}(f_i - f_j)^2 \tag{1}$$

**Graph Neural Networks** We consider GNNs that act in the graph spatial domain as message passing (Gilmer et al., 2017). We focus on the Graph Isomorphism Network (GIN) (Xu et al., 2018), which uses a SUM-aggregator on messages received by each node to achieve a better understanding of the graph structure:

$$\mathbf{x}_i^l = \mathbf{M}_\Theta^l \left( \mathbf{x}_i^{(l-1)} + \sum_{v_j \in \mathcal{N}(v_i)} A_{i,j}^{(l-1)} \mathbf{x}_j^{(l-1)} \right) \tag{2}$$

where $\mathbf{M}_\Theta^l$ is a neural network with trainable parameters $\Theta$, $x_i$ is the feature vector for node $v_i$, $v_j \in \mathcal{N}(v_i)$ are the neighbors of $v_i$ and $l$ is the layer number. Notice the term $A_{i,j}$ that takes into account the edge weight between nodes $v_i$ and $v_j$ when $A$ is not binary.

In this work, we focus mainly on molecular graphs in supervised settings where, given a molecule $m$ and its corresponding molecular graph $G_m$, we aim to predict some properties of $m$. Molecular graphs present two particularities: (1) they are often sparse and (2) there is no regularity in the graph signal (non-smooth variation) as adjacent nodes tend not to have similar features.

## 3.2 GRAPH DOWNSAMPLING

This section details how LaPool downsamples the original graph by selecting a set $V_C$ of nodes as centroids after $l$ consecutive GC layers.

**Centroid Selection** For any given vertex $v_i$, we can define a local measure $s_i$ of intensity of signal variation around $v_i$. As $s_i$ measures how different the signal residing at a node $v_i$ is from its neighbors, we are interested in the set $V_C$ of nodes that have an intensity of signal variation $s_i$ greater than their neighborhood. In this work, we use a definition of local signal variation similar to the *local normalized neighboring signal variation* described in (Chen et al., 2015b), with the only difference being the absence of degree normalization:

$$s_i = \Big\| \sum_{j \in \mathcal{N}(v_i)} A_{i,j}(\mathbf{x}_i - \mathbf{x}_j) \Big\|_2, \quad S = \big[s_1, \ldots s_n\big]^\intercal = \|LX\|_{2,\mathbf{R}^d} \quad V_C = \text{top}_V(LS, k) \quad (3)$$

where $\text{top}_V(L \cdot S, k)$ corresponds to the top $k$ nodes with the greatest intensity of signal variation, and where $\| \cdot \|_{2,\mathbf{R}^d}$ corresponds to taking the vector norms over the d-dimensional rows of $LX$. Instead of using the direct neighbors, one can also generalize the computation of $S$ in equation 3 to an $h-$hop neighborhood by taking powers of the Laplacian.

Observe that the GC layers preceding each pooling step perform a smoothing of the graph signal and thus act as a low-pass filter. Eq. (3) emphasizes instead the high variation regions, resulting overall in a filtering of $X$ that attenuates low and high-frequency noise, yet retains the important signal information. The intuition of using the Laplacian maxima for selecting the centroids is that a smooth signal can be very well approximated using a linear interpolation between its local maxima and minima. This is in contrast with most approaches in GSP that use the lower frequencies for signal conservation but requires the signal to be k-bandlimited (Ma et al., 2019; Chen et al., 2015c;a). For a 1D signal, LaPool selects points, usually near the maxima/minima, where the derivative changes the most and is hardest to interpolate linearly. For molecular graphs, this often corresponds to sampling a subset of nodes critical for reconstructing the original molecule.

**Dynamic Selection of the Centroids** The method presented in Eq. (3) implies the selection of $k$ centroids. Because the optimal value of $k$ can be graph-dependent and might result in densely located centroids, we explore alternative in which we dynamically select the nodes with signal variation $s_i$ greater than its neighbors $s_j$:

$$V_C = \{v_i \in V \mid \forall\, v_j, s_i - A_{ij}s_j > 0\} \quad (4)$$

## 3.3 LEARNING THE NODE-TO-CLUSTER ASSIGNMENT MATRIX

Once the set $V_C$ of centroid nodes is determined, we compute a mapping of the remaining "follower" nodes $V_F = V \setminus V_C$ into the new clusters formed by the nodes in $V_C$. This mapping gives the cluster assignment $C = \big[\mathbf{c}_1, \ldots \mathbf{c}_n\big]^\intercal \in [0,1]^{n \times m}$ s.t. $\forall i : \mathbf{1}\mathbf{c}_i^\intercal = 1$, where each row $\mathbf{c}_i$ corresponds to the affinity of node $v_i$ towards each of the $m$ clusters in $V_C$.

Let $X^{(l)}$ be the node embedding matrix at an arbitrary layer and $X_C^{(l)}$ the embedding of the "centroids". We compute $C$ using a soft-attention mechanism (Graves et al., 2014) measured by the cosine similarity between $X^{(l)}$ and $X_C^{(l)}$ :

$$\mathbf{c}_i = \begin{cases} \delta_{i,j} & \text{if } v_i \in V_C \\ \text{sparsemax}\Big(\beta_i \frac{\mathbf{x_i}^{(l)} \cdot X_C^{(l)}}{\|\mathbf{x_i}^{(l)}\|\|X_C^{(l)}\|}\Big) & \text{otherwise} \end{cases} \quad (5)$$

where $\delta_{i,j}$ is the Kronecker delta and sparsemax (Laha et al., 2016; Martins and Astudillo, 2016), is an alternative to the softmax operator defined by:

$\text{sparsemax}(\mathbf{z}) = \underset{\mathbf{p} \in \Delta^{K-1}}{\arg\min} \|\mathbf{p} - \mathbf{z}\|^2$ which corresponds to the euclidean projection of $\mathbf{z}$ onto the probability simplex $\Delta^{K-1} = \{\mathbf{p} \in \mathbb{R}^K \mid \mathbf{1}^\intercal \mathbf{p} = 1, \mathbf{p} \geq 0\}$. The sparsemax operator ensures the sparsity of the attention coefficients and encourages the assignment of each node to a single centroid. It further alleviates the need for entropy minimization as done in DiffPool.

Eq. (5) also prevents the selected centroid nodes from being assigned to other clusters. Moreover, notice the term $\beta_i$ that regularizes the value of the attention for each node. We can define $\beta_i = \frac{1}{\mathbf{d_{i,V_C}}}$, where $\mathbf{d_{i,V_C}}$ is the shortest path distance between each node $v_i \in V_F$ and centroids in $V_C$. Although this regularization incurs a cost $\mathcal{O}(|V|^2|V_C|)$, it will strengthen the affinity to closer centroids and ensure the connectivity of the resulting pooled graph. Note that this regularization is considered as a hyperparameter of the layer and can be turned off, or alternatively, the mapping can be restricted to centroids within a fixed $h-$hop neighborhood of each node.

Finally, after $C^l$ is computed at layer $l$, the coarsened graph $G^{(l+1)} = \langle V^{(l+1)}, A^{(l+1)}, X^{(l+1)} \rangle$ is computed using Eq. (6), as in (Ying et al., 2018). In these equations, $M_\Psi$ is a neural network with trainable parameters $\Psi$ that is used to update the embedding of nodes in $G^{(l+1)}$ after the mapping.

$$A^{(l+1)} = C^{(l)\mathsf{T}} A^{(l)} C^{(l)} \in \mathbb{R}^{|V_C^{(l)}| \times |V_C^{(l)}|}, \qquad X^{(l+1)} = M_\Psi\big(C^{(l)\mathsf{T}} X^{(l)}\big) \tag{6}$$

This process can be repeated by feeding the new graph $G^{(l+1)}$ into another GNN layer.

### 3.4 PROPERTIES OF THE LAPOOL METHOD

**Preservation of Structural Information** In addition to identifying graph nodes as centroids for pooling in a data-driven way, LaPool retains the feature content of the other nodes in a graph via the soft-assignment of followers to their centroids.

**Substructure Identification** By construction, the soft assignment of nodes to centroids clusters *existing* substructures of the graph together, thus identifying important subgraphs according to the classification task. By controlling for differences in signal variations within neighborhoods, we encourage these clusters to be spread out across different areas of the graph.

**Dynamic Cluster Dimension** As discussed in section 3.2, LaPool offers the unique flexibility of determining the clustering dynamically, when training graphs sequentially, or statically when performing batch training.

**Permutation Invariance** It is trivial to show that LaPool is permutation invariant as long as the GNN used as its basis is permutation invariant, since both the graph downsampling (Eq. 3,4) and the node mapping (Eq. 5,6) are not affected by any permutation on the vertex set.

**Emphasizing the *Strong* Features** Similar to how most CNNs implement a max-pooling layer to emphasize the *strong* features, LaPool does so by selecting the nodes with high signal as centroids. For molecular graphs, the centroids are biased towards high degree nodes and atoms different than their neighbors (e.g. a Nitrogen in a Carbon ring).

## 4 RESULTS AND DISCUSSION

A fundamental objective of LaPool is to learn an interpretable representation of sparse graphs, notably molecular substructures. We argue that this is an essential step towards shedding light upon the decision process within neural networks and ultimately increasing their utility in the design of new drug-like molecules. This implies that GNNs should be able to identify semantically important substructure components from molecular graphs, and eventually reconstruct these graphs from such components. This stems from the intuition that molecular validity and functional properties derive more from chemical fragments than individual atoms.

Our experimental results thus aim to empirically demonstrate the following properties of LaPool, as benchmarked against current state-of-the-art pooling models and the Graph Isomorphism Network:

- LaPool's consideration of semantically important information such as node distance translates to improved performance on molecular understanding and molecular activity prediction tasks.
- Visualization of LaPool's behaviour at the pooling layer demonstrates its ability to identify coherent and meaningful molecular substructures.
- The hierarchical representation enforced by LaPool, which preserves the original graph structure improves model interpretability.

- Learning meaningful substructures can be leveraged to construct a generative model which leads to more realistic and feasible molecules.

Throughout our experiments, we use the same architecture for all models to ensure an even comparison across all pooling layers: 2 layer GNNs with 128 filters each before the optional pooling layer, followed by 2 GNNs with 64 filters each and two fully connected layers including the output layer. Detailed information on architectural tuning and pooling-specific hyper-parameter search are provided in Supplemental Section A.

## 4.1 BENCHMARK ON MOLECULAR GRAPH UNDERSTANDING

DiffPool and Graph U-Net models have been shown to outperform standard graph convolution networks on several graph benchmark datasets (Ying et al., 2018; Gao and Ji, 2018). Although not explicitly stated, both methods are most effective when the graph signal is smooth. In such cases where adjacent nodes tend to be similar, the DiffPool procedure will cluster together nodes in the same neighborhood, maintaining the overall graph structure, while Graph U-net will select nodes in the same neighborhood and will not create isolated components that no longer exchange information. On molecular graphs, however, the graph signal is rarely smooth. Therefore, we expect these two methods to be less effective at identifying the important molecular substructures, given that they do not explicitly consider structural relationships. We demonstrate this empirically by extracting known molecular substructure information from publicly available molecular datasets and evaluating performance in identifying these structures. We use a subset of approximately 17,000 molecules extracted from the ChEMBL database (Li et al., 2018c) and benchmark all methods on different types of substructures to verify the robustness of the comparison.

As shown in Table 1, the capture of structural relationships translates into superior performance of LaPool, as measured across standard metrics on various substructure prediction tasks. Indeed, we find that for predicting the presence of both 86 molecular fragments arising purely from structural information, as well as 55 structural alerts associated with molecule toxicity, LaPool globally outperforms other pooling models and GIN for the F1 and ROC-AUC metrics (micro-averaged to deal with high class imbalance). In particular, on the harder and extremely imbalanced molecular alerts prediction task, all models performed poorly compared to LaPool, suggesting that the hierarchical representation learned by LaPool helps to achieve a better understanding of the molecular graphs.

**Table 1:** Molecular fragment prediction results on ChEMBL dataset (5-fold cross-validation).

|  | Fragments | | Alerts | |
|---|---|---|---|---|
|  | F1 | ROC-AUC | F1 | ROC-AUC |
| GIN | **99.436 $\pm$ 0.545** | **99.991 $\pm$ 0.004** | 31.759 $\pm$ 3.728 | 82.495 $\pm$ 8.429 |
| DiffPool | 97.961 $\pm$ 0.384 | 99.967 $\pm$ 0.025 | 48.638 $\pm$ 9.916 | 76.537 $\pm$ 0.241 |
| Graph U-net | 95.469 $\pm$ 1.414 | 99.962 $\pm$ 0.033 | 37.585 $\pm$ 2.978 | 85.124 $\pm$ 8.447 |
| LaPool | 98.980 $\pm$ 0.506 | **99.994 $\pm$ 0.000** | **78.592 $\pm$ 7.217** | **94.164 $\pm$ 1.784** |

## 4.2 EXPERIMENTS ON STANDARD GRAPH CLASSIFICATION BENCHMARKS

In addition to evaluating molecular structural understanding of the pooling models, we benchmark our model on molecular toxicity prediction using the TOX21 dataset (Council et al., 2007). We further conduct experiments on non-molecular benchmark graph datasets (DD, PROTEINS, FRANKEN-STEIN), which usually contain larger and often denser graphs compared to molecular graphs (see Supplemental section C for dataset statistics). For TOX21, we report the test ROC-AUC averaged over 5-folds (following the 80-10-10 split proportion used in Wu et al. (2018)), while we follow prior work (Ying et al., 2018) on the remaining datasets by reporting the best accuracy on a 10-fold cross-validation.

As shown in Table 2, LaPool outperforms all other approaches on the well known TOX21 dataset and on the PROTEINS and FRANKENSTEIN, both of which contain non-molecular graphs with size similar to the TOX21 molecules. In particular, on the PROTEINS dataset LaPool achieved an

accuracy of 83.83, representing a significant gap relative to DiffPool, its closest competitor, at 77.25. This suggests that the LaPool method is not restricted to molecular data but has broad applicability, especially in the context of sparse graph classification.

**Table 2:** Performance evaluation on standard benchmark graphs

|            | TOX21           | DD        | PROTEINS  | FRANKENSTEIN |
|------------|-----------------|-----------|-----------|--------------|
| GIN        | $82.90 \pm 0.69$ | 77.97     | 66.47     | 68.20        |
| DiffPool   | $82.37 \pm 0.90$ | **85.88** | 77.25     | 69.12        |
| Graph U-net | $81.41 \pm 0.60$ | 77.40     | 74.50     | 66.16        |
| LaPool     | **$83.42 \pm 0.97$** | 81.36 | **83.83** | **69.74**    |

### 4.3 MOLECULAR GENERATION

We aim to showcase LaPool's utility in drug discovery by demonstrating that it can be leveraged to generate molecules. In previous work, GANs and VAEs were used to generate either string or graph representations of molecules. Here, we use the GAN-based Wasserstein Auto-Encoder recently proposed in (Tolstikhin et al., 2017) to model the data distribution of molecules (see Figure 1 in Supplemental Material). For the encoder, we use a similar network architecture as in our supervised experiments. The decoder and discriminator are simple MLPs, with complete architectural details provided in Supplemental Section A.4. Although the encoder is permutation invariant, the decoding process may not be. To force the decoder to learn a single graph ordering, we use a canonicalization algorithm (Schneider et al., 2015) that reorders atoms to ensure a unique graph for each molecule. We further improve the robustness of our generative model to node permutations by computing the reconstruction loss using a permutation-invariant embedding, parameterized by a GIN, on both the input and reconstructed graphs (see Supplemental Section A.4.2). We find that such a formulation improves the reconstruction loss and increases the ratio of valid molecules generated.

**Dataset and Baseline Models** Following previous work on molecular generation, we evaluate our generative model with an encoder enhanced by the LaPool layer (referred to as WAE-LaP) on the QM9 molecular dataset (Ramakrishnan et al., 2014). This dataset contains 133,885 small drug-like organic compounds with up to 9 heavy atoms (C, O, N, F). We compare WAE-LaP to alternatives within our WAE framework where either no pooling is used (WAE-GNN) or where DiffPool is used as the pooling layer (WAE-Diff). Our results are also compared to previous results on the same dataset, including Grammar-VAE, GraphVAE, and MolGAN.

**Evaluation Metrics** We measure the performance of the generative model using metrics standard in the field: validity (proportion of valid molecules from generated samples), uniqueness (proportion of unique molecules generated), and novelty (proportion of generated samples not found in the training set). All metrics were computed on a set of 10,000 generated molecules.

**Table 3:** Performance comparison of the generative models on QM9. Values are reported in percentages and baseline results are taken from (De Cao and Kipf, 2018).

|          | WAE-GNN | WAE-Diff | WAE-LaP | Grammar-VAE | GraphVAE | MolGAN |
|----------|---------|----------|---------|-------------|----------|--------|
| % Valid  | 96.8    | 97.2     | **98.8** | 60.2        | 91.0     | 98.1   |
| % Unique | 50.0    | 29.3     | **65.5** | 9.3         | 24.1     | 10.4   |
| % Novel  | 78.9    | 78.9     | 78.4    | 80.9        | 61.0     | **94.2** |

As shown in Table 3, WAE-LaP generated the most valid and unique molecules compared to all WAE-based generative models with slightly lower but similar novelty. Although MolGAN performed best on the novelty metric, it has among the lowest percentage of unique molecules. We hypothesize that the decrease in novelty observed with LaPool might be a result of its pooling mechanism encouraging fragment novelty during sampling, thus limiting novelty resulting from rearrangement at the atom level. Nevertheless, as all WAE-based methods produced similar proportions of novel molecules, our results still suggest that combining LaPool with other generative approaches could improve the uniqueness and validity of generated compounds. We therefore conclude that the pooling performed by LaPool can improve molecular graph representation, which is crucial in a generative setting.

## 4.4 IMPROVED INTERPRETABILITY

To better understand the insights provided by LaPool, we conduct model interpretability experiments on molecular fragment prediction. Here we refer to interpretability as the degree to which a human (in this context, a medicinal chemist) can understand the cause of the model's decision (Miller, 2018). This explains our focus on fragment prediction since in that setting an "interpretable model" that achieves high performance would need to first understand the graph structure and the relationship between nodes.

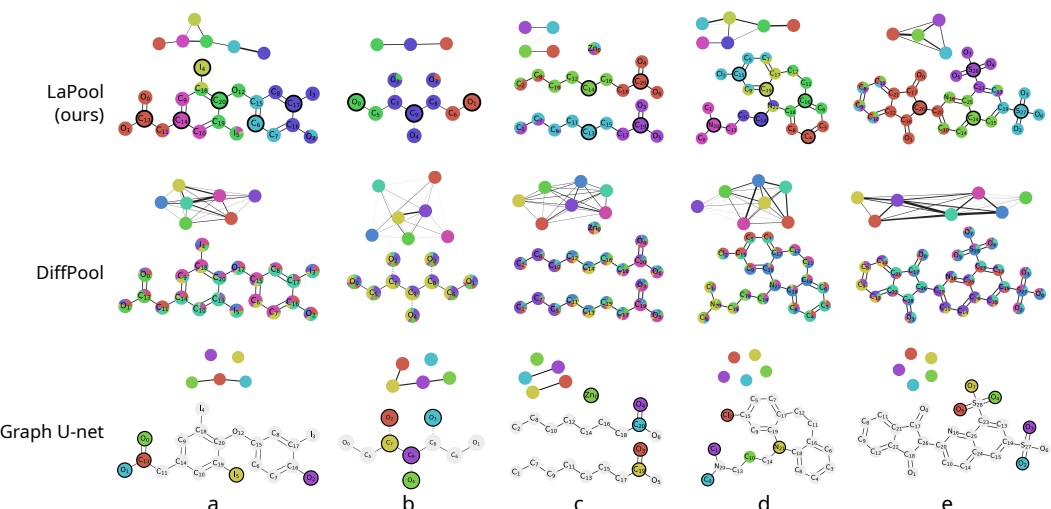

**Figure 2:** Visualization of node selection and/or clustering performed by DiffPool, Graph U-net and LaPool for structural alert prediction. Dynamic clustering was used for LaPool, while the cluster size ($k$) resulting in the the best model was used for DiffPool($k = 7$) and Graph U-net ($k = 5$). The top graph is the pooled graph. The bottom graph is the original molecule, with the pie-charts representing the cluster affinity of each node. For LaPool and Graph U-net, the bold nodes respectively represent the chosen centroids and selected nodes.

We first investigate the behavior of LaPool and DiffPool by analyzing the clustering made at the pooling layer level. This comparison was limited to DiffPool since Graph U-net does not perform a node clustering. We argue that an interpretable pooling layer should preserve the overall graph structure after pooling and produce *meaningful* clusters that could provide insight into the contribution of each molecular subgraph from the perspective of an expert chemist. While defining what is meaningful is inherently subjective, we attempt to shed light on these models by observing their behavior in the drug discovery domain, using our understanding of chemical structure as reference.

We show in Figure 2 that LaPool is able to coarsen the molecular graphs into sparsely connected graphs, which can be interpreted as the *skeleton* of the molecules. Indeed, the data-driven dynamic segmentation it performed is akin to chemical fragmentation (Gordon et al., 2011). In contrast, DiffPool's cluster assignment is more uniform across the graph, leading to densely connected coarsened graphs which are less interpretable from a chemical viewpoint. In particular, it fails in the presence of molecular symmetry, as it encourages the mapping of nodes with similar features to the same clusters. This is illustrated in both example (c) which shows how DiffPool creates a fully connected graph from an originally disconnected graph, and example (b) which shows how symmetric elements, despite being far from each other, are assigned identically. On the other hand, we observe that Graph U-net ignores the graph structure, typically disconnecting it. It also appears very biased toward selecting atoms in similar environment to ones already selected. Such failures are not present when using LaPool, since the dynamic centroid selection and the subsequent distance regularization enforce preservation of the molecular graph structure. A typical failure case for LaPool is seen in (e) and corresponds to a missing centroid node in a given region of the graph, which results in a soft assignment of the region to multiple clusters. However, this behavior is inherent to most DiffPool samples since the fixed number of clusters and the inability to consider node distance cannot account for the diversity of molecular datasets.

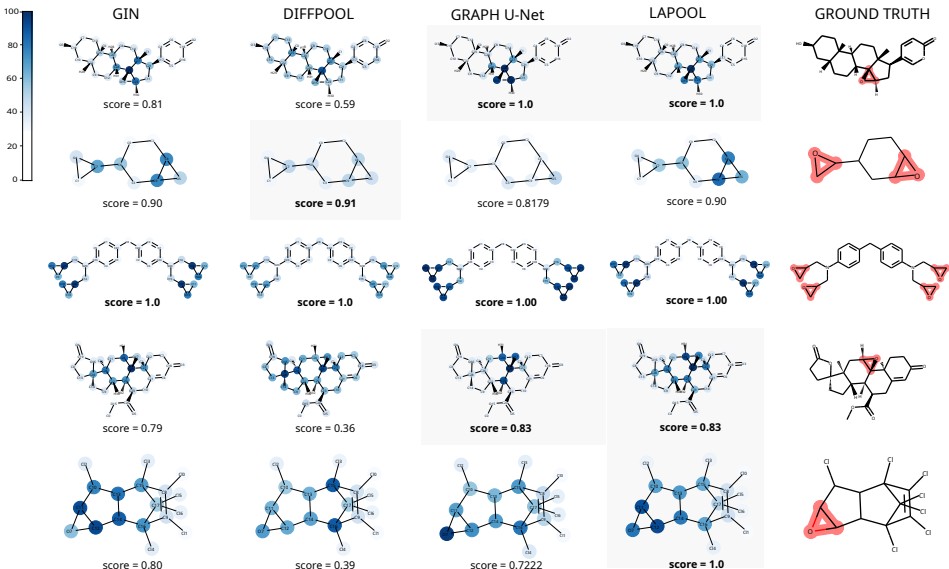

**Figure 3:** Example of node importance for GIN, DiffPool, Graph U-Net and LaPool models. The ground truth is shown on the right and nodes are highlighted according to their predicted importance. The computed interpretability score is indicated below each molecules and the best performing models on each example are highlighted in gray. The computed interpretability score is shown for each model.

In addition to assessing the quality of clustering performed by LaPool and DiffPool, we attempt to directly target interpretability by computing an explanation of each model decision and comparing it to a ground truth. We design a simple experiment in which we predict the presence of either Epoxide, Thioepoxide, or Aziridine substructures (denoted by the molecular pattern "C1[O,S,N]C1"), that are indicative of molecular toxicity. Interpretability is therefore defined as the accuracy of the importance attributed by each model to relevant substructures of the input molecules, given the presence of the underlying ground truth fragment we wish to predict. Similar to (Pope et al., 2019), we adapt an existing explainability method for CNNs to GNNs. Specifically, we choose to compute the integrated gradient (Sundararajan et al., 2017) over the input node features due to its stability and robustness in the presence of zero-value features (see Supplemental section C for discussion and alternate approach). Next, we derive an importance score for each node using the L1-norm of the feature attribution map for the node. By both qualitatively observing samples from the data (Figure 3), and by measuring the PR-AUC over the computed importance values given the ground truth to assess the ability to distinguish between important and non-important nodes (Table 4), we find LaPool to more robustly identify the salient structure, resulting in improved overall interpretability. An interesting outcome of this experiment is the performance of Graph U-net, ranked second best. Such performance is a direct result of using a cluster size large enough to cover the toxic fragment size.

**Table 4:** Comparison of prediction explainability based on average PR-AUC over node attribution

|  | GIN | DiffPool | Graph U-Net | LaPool |
|---|---|---|---|---|
| avg. PR-AUC | 0.876 | 0.799 | 0.879 | **0.906** |

## 5 CONCLUSION

In this work, we have proposed LaPool, a differentiable and robust pooling operator for molecular and sparse graphs that considers both node information and graph structure. In doing so, we have proposed a method which is able to identify important substructures of a graph by leveraging the graph Laplacian. In contrast with previous work, this method retains the connectivity structure and

feature information of the graph during the coarsening procedure, while encouraging nodes belonging to the same substructure to be mapped together in the coarsened graph.

Incorporating the proposed pooling layer into existing graph neural networks, we have demonstrated that the enforced hierarchization allows for the capture of a richer and more relevant set of features at the graph-level representation. We discussed the performance of LaPool relative to existing graph pooling layers and demonstrated on both molecular graph classification and generation benchmarks that LaPool outperforms existing graph pooling modules and produces more interpretable results. In particular, we argue that the molecular graph segmentation performed by LaPool provides greater insight into molecular activity and that the associated properties can be leveraged in drug discovery. Finally, we show that although LaPool was designed for molecular graphs, it generalizes well to other graph types. In future work, we aim to investigate how additional sources of information such as edge features could be incorporated into the graph pooling process.

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

# A    NETWORK ARCHITECTURE AND TRAINING PROCEDURE

Below, we describe the network architecture and the training procedure used for both supervised and generative experiments.

## A.1    EDGE ATTRIBUTES

Part of the work presented assumes the absence of edge attributes in the graphs. However, in molecular graphs, the nature of a bond between two atoms plays an important role regarding activity and property. As such, edge types should be considered, especially in generative models. To consider this, we add to our network an initial Edge-GC layer described in the following.

Let $G = \langle V, E, X \rangle$ be an undirected molecular graph, such that $E = [E_1, \ldots E_k] \in \{0, 1\}^{e \times n \times n}$ where $n$ is the number of nodes in the graph and $e$ is the number of possible edge. We have that

$$\sum_{1 \leq i \leq e} E_{::i} = A \in \{0, 1\}^{n \times n} \tag{7}$$

where $A$ is the adjacency matrix of the graph. The Edge GC layer is simply defined as :

$$Y = M_{\Theta_1}(E_1, X) \| \ldots \| M_{\Theta_e}(E_e, X) \tag{8}$$

where $\|$ is the concatenation operator on the node feature dimension and $M_{\Theta_1}$ are graph neural networks parameterized to learn different features for each edge type. A new graph defined as $G' = \langle V, A, Y \rangle$ can then be feed into the subsequent layers of the network.

## A.2    ATOM AND EDGE FEATURES

In our experiments, the initial node feature tensor is represented by a one-hot encoding of atoms (ignoring hydrogens) within the respective datasets and additional properties such as the atom implicit valence, its formal charge, number of radical electrons and whether it is in a molecular ring. For edge attributes, we consider the single, double and triple bond, which were enough to cover all molecules in the datasets, given that feature extraction was preceded by kekulization of molecules.

## A.3    SUPERVISED EXPERIMENTS

In all of our supervised experiments, we use a graph convolution module consisting of two graph convolutional layers of 128 channels each with ReLU activation; followed by an optional hierarchical graph pooling layer; then two additional graph convolution layers (64) with skip connection to introduce jumping knowledge and a gated global graph pooling layer (Li et al., 2015) to yield a graph-level representation. This is further followed by one fully connected layers (128) with batch normalization and ReLU activation, finalized by a linear output layer with appropriate activation for the task readouts. Notice that we used one pooling layer, since no noticeable improvement was observed when using more in our experimental setting.

For DiffPool, we performed a hyperparameter search to find the optimal number of clusters (12.5%, 25%, 50% of the maximum number of nodes in the batch (Ying et al., 2018)). A similar search is also performed for the Graph U-net pooling layer. For LaPool, we consider the same number of clusters and the dynamic node seelction. We also performed a grid search over the window size $k$ used as regularization to prevent nodes from mapping to centroids that are more than $k$-hop away as an alternative to the distance-regularized version. The grid search was performed for $k \in \{1, 2, 3\}$.

For the supervised experiments, we use a batch size of 64 and train the networks for 100 epochs, with early stopping.

## A.4    GENERATIVE MODELS

### A.4.1    WAE MODEL

We use a Wasserstein Auto-Encoder (WAE) as our generative model (see Figure S1. The WAE minimizes a penalized form of the Wasserstein distance between a model distribution and a target

**Figure S1:** Model architecture for the generative model. (a) We use a WAE, in which a generator (auto-encoder) progressively learns the true molecular data distribution. (b) Architecture used for the encoder network.

distribution, and has been shown to improve learning stability. As described in (Tolstikhin et al., 2017), we aim to minimize the following objective:

$$\inf_{Q(Z|X)\in\mathcal{Q}} \mathbb{E}_{P_X}\mathbb{E}_{Q(Z|X)}[cost(X, G(Z)] + \lambda\mathcal{D}_Z(Q_Z, P_Z) \tag{9}$$

where $\mathcal{Q}$ is any nonparametric set of probabilistic encoders, $D_Z$ is the Jensen-Shannon divergence between the learned latent distribution $Q_Z$ and prior $P_Z$, and $\lambda > 0$ is a hyperparameter. $D_Z$ is estimated using adversarial training (discriminator).

For our generative model, the encoder follows a similar structure as the network used for our supervised experiments, with the exception being that the network now learns a continuous latent space $q_\Psi(z|\mathcal{G})$ given a set of input molecular graphs $\mathcal{G} = \{G_1, \cdots, G_n\}$. More precisely, it consists of one edge graph layer, followed by two GCs (32 channels each), an optional hierarchical graph pooling, two additional GC layers (64, 64), then one global sum pooling step (128) and two fully connected layers (128), meaning the molecular graphs are embedded into a latent space of dimension 128. Instead of modeling the node/edge decoding with an autoregressive framework as done in recent works (You et al., 2018b; Assouel et al., 2018; Li et al., 2018d) to capture the interdependency between them, we used a simple MLP that takes the latent code $z$ as input an pass it through two fully connected layers (128, 64). The output of those layers is used as shared embedding for two networks: one predicting the upper triangular entries of the edge tensor, and the second predicting the node features tensor. This results in faster convergence.

For the discriminator, we use a simple MLP that predicts whether the latent code comes from a normal prior distribution $z \mathcal{N}(0, 1)$. This MLP is constituted by two stacked FCLs (64, 32) followed by an output layer with sigmoid activation. As in (Kadurin et al., 2017), we do not use batch-normalization, since it resulted in a mismatch between the discriminator and the generator.

All models use the same basic generative architecture, with the only difference being the presence of a pooling-layer and its associated parameters. For DiffPool, we fixed the number of cluster to three, while for LaPool, we use the distance-based regularization.

### A.4.2 Reconstruction loss

For each input molecular graph $G = \langle V, E, X \rangle$, the decoder reconstruct a graph $\tilde{G} = \langle \tilde{V}, \tilde{E}, \tilde{X} \rangle$. Since we use a canonical ordering (available in RDKit) to construct $G$ from the SMILES representation of molecules, the decoder is forced to learn how to generate a graph under this order. Therefore, the decoding process is not necessarily able to consider permutations on the vertices set, and generation of isomorphic graphs will be heavily penalized in the reconstruction loss. In (Simonovsky and Komodakis), the authors use an expensive graph matching procedure to overcome that limitation. We argue that it suffices to compute the reconstruction loss on $\gamma(G)$ and $\gamma(\tilde{G})$, where $\gamma$ is a permutation invariant embedding function. As a heuristic, we used a Graph Isomorphism Network (GIN), with weights fixed to 1, in order to approximate the Weisfeiler-Lehman graph isomorphism test (see (Xu

et al., 2018) for more details). In particular, we use an edge-aware GIN layer (see section A.1) to embed both $G$ and $\tilde{G}$. The reconstruction loss is then defined as:

$$\mathcal{L}_{rec} = \frac{1}{|V|} \sum_i (\gamma(G)_i - \gamma(\tilde{G})_i)^2 \tag{10}$$

Our experiments show that this loss function was able to produce a higher number of valid molecules, although we speculate that such a heuristic might prove harder to optimize on datasets with larger graphs.

### A.4.3 TRAINING PROCEDURE

The QM9 dataset was split into a train (60%), valid (20%) and a hold-out test dataset (20%). Note that only 25% of the training set is sampled during each epoch (batch size 32). The generator network (encoder-decoder) and the discriminator network are trained independently, using the Adam optimizer Kingma and Ba (2014) with an initial learning rate of $1e - 4$ for the generator and $1e - 3$ for the discriminator. During training, we slowly reduce the learning rate by a factor of 0.5, for the generator, on plateau. To stabilize the learning process and prevent the discriminator from becoming "too good" at distinguishing the true data distribution from the prior, we train the generator two times more often.

## B GENERATED MOLECULES

Below, we highlight a few molecules generated by WAE-LaP on the QM9 dataset.

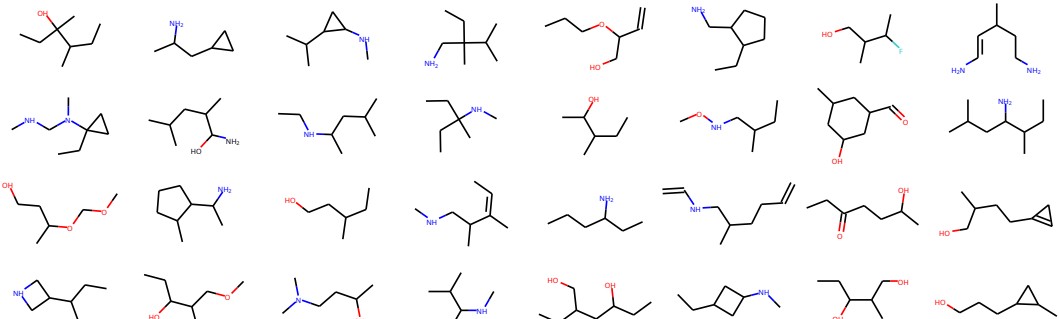

**Figure S2:** Example of molecules generated by WAE-LaP. Hydrogen atoms are not shown for simplicity.

## C DATASET STATISTICS

**Table 5:** Dataset statistics and properties.

|            | TOX21 | DD     | PROTEINS | FRANKENSTEIN |
|------------|-------|--------|----------|--------------|
| Avg. nodes | 18.51 | 284.32 | 39.05    | 16.83        |
| Avg. edges | 19.23 | 715.66 | 72.82    | 17.88        |
| #Graphs    | 8014  | 1178   | 1113     | 4337         |
| #Classes   | 12    | 2      | 2        | 2            |

## D NODE IMPORTANCE INTERPRETABILITY SCORE

In addition to assessing the quality of clustering performed by LaPool and DiffPool, we attempt to measure the interpretability of their predictions. We consider a setting in which the goal is to predict the presence of either Epoxides, Thioepoxides or Aziridines substructures in molecular graphs. These

three fragments correspond to structural alerts that are often indicative of molecular toxicity. We attempt to identify the relational inductive bias used by each model during prediction. In our setting, we define the interpretability of a model as its ability to focus on nodes that are directly relevant to the structural alerts and leverage that information for its prediction. In other words, we expect the most important nodes for the model prediction to correspond to nodes that are part of the structural alerts. We measure the importance of each atom toward the model prediction using the Integrated gradient method. Briefly, we compute perturbation of node and edge attributes over a continuous spectrum, then integrate the gradient of each of the model loss with respect to both the perturbed adjacency matrix and node features. Similar to saliency maps, we then take the sum of the absolute integrated gradients over each node as an approximate attribution score for the nodes. Finally, we compute the interpretability score using the Precision-Recall AUC between measured importance and ground truth which is defined by a binary mask of nodes that are part of the structural alerts. The PR-AUC allows us to assess the node importance separation capacity of each model while taking imbalance into account. We only focus on positive predictions for each model. As an alternative to the Integrated Gradient, we also measure the interpretability score using Guided BackPropagation (see Table 6)

**Table 6:** Comparison of prediction explainability based on average PR-AUC over node attribution using various explainability framework.

|  | GIN | DiffPool | Graph U-net | LaPool |
|---|---|---|---|---|
| Integrated Gradient | 0.876 | 0.799 | 0.879 | **0.906** |
| Guided BackPropagation | 0.819 | 0.739 | 0.835 | **0.857** |

## E  SIGNAL PRESERVATION THROUGH LAPLACIAN MAXIMA

We illustrate here on a 1-d signal $S$, how using the Laplacian maxima serves to retain the most prominent regions of the graph signal, after smoothing (Figure S3). We measure the energy conservation after downsampling: $\delta_E(S) = E(S) - E(S_{down})$ of the 1-d signal energy to highlight why selecting the Laplacian maxima allow reconstructing the signal with a low error when compared to the minimum Laplacian (which focuses on low frequencies). The energy $E_S$ of a discrete signal $y_i$ is defined in (11), and is similar to the energy of a wave in a physical system (without the constants).

$$E_S = \sum_i |y_i|^2 \tag{11}$$

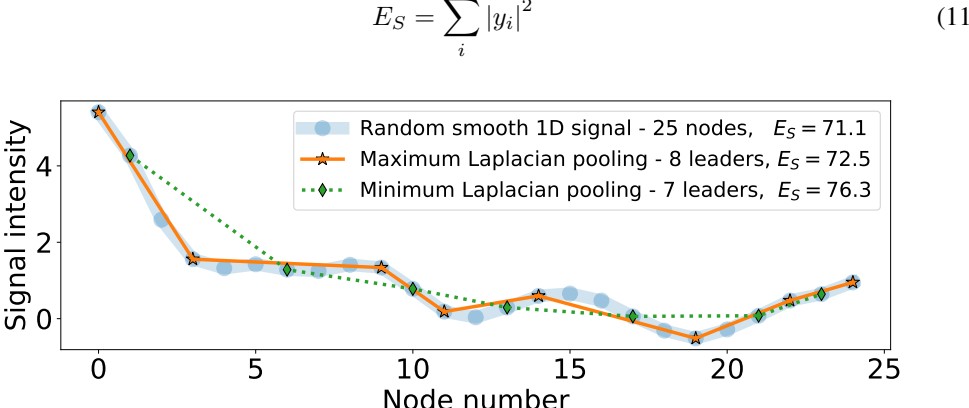

**Figure S3:** Comparison of maximum/minimum Laplacian pooling for a random and smoothed signal on a 1D graph with 25 nodes. The graph energy $E_S$ is indicated.

To mimic the molecular graph signal at the pooling stage, the given signal is built from an 8-terms random Fourier series with added Gaussian noise, then smoothed with 2 consecutive neighbor average smoothing. For the pooling methods, a linear interpolation is used to cover the same signal space before computing $E_S$. As expected, the maxima Laplacian selection requires a fewer number of samples for signal reconstruction and energy preservaton. It also significantly outperforms minima selection.

