# OpenReview forum: "Towards Interpretable Molecular Graph Representation Learning"
_ICLR.cc/2020/Conference — Reject_

### Official Review · AnonReviewer1 · 2019-10-21
**Official Blind Review #1**

**Rating:** 1

**Review:**

This paper proposes a graph down-sampling component named LaPool. It uses the graph signal to dynamically select some "important" centroids and learn a sparse assignment matrix for clustering the remaining nodes.

This paper should be rejected due to the following reasons.

1. Strange new results compared with its previous version

The paper has a previous arxiv version. While the method does not change, the performance in this current submission has dramatic change compared with previous version: the proposed model seems much improved, while some important baselines (that outperform the proposed model) only have 50% of it previous performance in this submission.

For example, for Table 1 in the current version, the proposed model reached almost perfect scores in both F1 and AUC. However in previous version for the same setting and experiment (in Table 2 and 3 of its arxiv version), the performance are much lower especially the Structural alert prediction results.  Their previous results, which show the model does not perform better than DiffPool .

For the structural alert prediction results on DiffPool, below I copied and pasted the results from the arxiv version


Table 3: Structural alert prediction results
------------------------------------------------------------------------------------------------------------
                                             Tox21                   |               ChEMBL
                       F1-macro F1-micro ROC-AUC F1-macro F1-micro ROC-AUC
------------------------------------------------------------------------------------------------------------
GIN                     78.9        68.3             72.6           93.6         76.7          59.2
DiffPool              79.2        68.0             75.6           94.5        83.3          59.3
Graph U-net      71.1        47.6             67.9           92.9        68.1          59.3
LaPooldistance 80.6        74.2             73.5           95.2        81.3          59.5
LaPoolunreg     81.3        72.8             74.1           94.1        75.8          58.9
LaPool3hop       79.1        71.6             74.8           93.8        75.0          59.1
------------------------------------------------------------------------------------------------------------
One example:
(1) in the ICLR version, The F1 score for DiffPool is only 48.638 ± 9.916 on ChEMBL data (about 50% of its previous level) but F1 for the proposed method is improved. Why is that?
(2) same as baseline  Graph U-net , the ICLR version reports F1 37.585 ± 2.978, why is that?
(3) same as GIN, in the ICLR version, F1 is only 31.759 ± 3.728, less than 50% of its previous level in arxiv version.

The author needs to justify this dramatic change.

2. Although LaPool can dynamically select centroids, for a dense graph such as a complete graph, only one centroid will remain there since there is only one node which has larger signal variation than all its neighbors, as shown in Eq. 4. This consequently hurts the model performance on more dense graphs. That may be why in Table 2, on dense data "DD", LaPool performs much worse than baseline DiffPool. Also on another dense data "FRANKENSTEIN", LaPool does not performs significantly better than DiffPool.


3, the evaluation of interpretability is not convincing. This paper considers  "interpretability as the degree to which a human (in this context, a medicinal chemist) can understand the cause of the model’s decision". Therefore, the conclusion that this model is more interpretable is based on only one person's subjective judgement.  Even so, from the scores the model does not outperform baseline "GIN" that much.


**Experience Assessment:**

I have published one or two papers in this area.

**Review Assessment: Checking Correctness Of Derivations And Theory:**

I assessed the sensibility of the derivations and theory.

**Review Assessment: Checking Correctness Of Experiments:**

I carefully checked the experiments.

**Review Assessment: Thoroughness In Paper Reading:**

I read the paper thoroughly.

---

> ### Author Response · Authors · 2019-11-09
> **Response to reviewer #1**
>
> We thank the reviewer for assessing our manuscript but do not agree with their conclusions.
>
> 1. First, we would like to confirm that we updated our experiments on the supervised benchmarks (due to a previously unseen bug in the one hot encoding of node type for the ChEMBL molecular graphs) which made all atoms to be of the same type and thus only considered graph structure.
>
> Unfortunately, it is not possible to remove the record of a paper on arxiv, but we believe this submission represents an improved and more accurate comparison of the methods, and in the spirit of scientific advancement should be the only one under consideration.
>
> Further, we do not agree with the reviewer's comment about  "important baselines (that outperform the proposed model)" dropping in performance, while the performance of our model "seems much improved" and would like to point out that performance has increased for *ALL* methods evaluated on *ALL* datasets.
>
> The reviewer also appears to ignore reported results on TOX21 and FRAGMENTS, while cherry-picking the previously reported F1-macro on ALERTS. In addition, the reviewer does not mention that on the ALERTS dataset the ROC-AUC of all methods has improved from almost random (0.6) to ~0.8. While it is indeed true that the F1-score decreases, it does so for ALL methods (see 81.3 vs. 78.59 on average for LaPool).
>
> In the case of the structural alerts task, the major difference is due to an updated splitting approach compared to the previous iteration (to deal with extreme class imbalance). See the RandomStratifiedSplitter at https://deepchem.io/docs/_modules/deepchem/splits/splitters.html. Indeed, none of the 55 standard alerts considered are present in more than 0.7% of the 17k molecules, meaning that simply predicting the largest class would yield high accuracy. As the ROC-AUC is undefined when no positive example is present for a given task within a batch, the previous iteration defaulted to a score of 0.5 in this case and underestimated the model performances. The stratified splitter addresses this issue. The ability of ALL models to separate positive and negative class has improved dramatically (see ROC-AUC). However, at a standard decision threshold of 0.5 to compute the F1-score, the baseline fails to predict the positive class, in contrast to LaPool. It is for this reason that we are reporting both "AUC" and "F1-score".
>
> 2. We would first like to emphasize that this method was developed primarily for molecular graphs, which are sparse in nature and thus pose a unique set of challenges. Nonetheless, we have demonstrated empirically that Lapool produces competitive results on other benchmarks and may therefore be useful in more general settings.
>
> On the reviewer’s comment about performance on complete graphs, we would like to point out that this is a degenerate case for almost any message-passing network which also aggregates node neighbours. Indeed, for most GNNs, all node features would be identical after the first information propagation, rendering the meaning of message passing algorithms on complete graphs useless.
>
> Regarding the reviewer’s comment on our performance on "FRANKENSTEIN" relative to DiffPool, we would like to emphasize that FRANKENSTEIN is a notoriously difficult benchmark on which all models usually perform similarly (and poorly).  To this end, our reported performance is indeed better than values reported for any of the models in this recent paper (https://arxiv.org/pdf/1904.08082.pdf ), which also include baselines such as DIFFPOOL and GRAPH U-Net (60~62% vs. 66-69% in our manuscript).
>
> 3. Regarding interpretability, we acknowledge the inherent challenges of subjectivity in this emerging field. As such, we have attempted to take a principled approach. We use a rather well-accepted definition of interpretability in ML (see Miller, 2018) as a guiding principle. Further, we believe it is entirely reasonable to define interpretability based on the viewpoint of experts in the field.
>
> In addition, we would like to emphasize that we have performed both qualitative (regarding node attribution, clustering quality, graph downsampling structure preservation) and quantitative assessments of interpretability in the context of molecular representations. Given that there is no universal measure of interpretability, how it is measured is inherently subjective and will in part depend on the task of interest. If the reviewer has specific interpretability experiments in mind for molecular graphs, we would be happy to run them.

---

> > ### Comment · AnonReviewer1 · 2019-11-09
> > **Dubious results that were not well explained**
> >
> > To AC and all reviewers: With respect, after reading the authors’ clarification, I still find the results very dubious and probably non-reproducible. In particular, the authors claim that all reported results of all tested methods were increased for all method but that factually contradicted with what i had pointed out in my original review. There seems to be a serious issue with the integrity of the reported result of this paper. To maintain the quality of ICLR, I hope we can all have a closer look into this case and in particular, the authors’ response. I am not convinced at all by the authors’ response.

---

> > > ### Author Response · Authors · 2019-11-10
> > > **The reviewer has an obvious bias against our work and does not appear to have compared results as claimed.**
> > >
> > > Frankly, this is ridiculous and offensive. We invite the AC and other reviewers to compare the previous results to the results in this submission, and urge them to read our explanations and draw their own conclusions. Further, our source code is public, and we will share not only exact running configurations but also trained models with anyone wishing to reproduce our results.
> > >
> > > It is apparent to us that Reviewer 1 either did not take the time needed to thoroughly compare the results (as claimed), or does not seem to have a strong grasp of F1-score and ROC-AUC. Moreover, this reviewer appears to have a strong bias against our work, conveniently (and unfairly) cherry-picking the difference in F1-score on a single dataset (ALERTS) in a bizarre attempt to discredit our results.
> > >
> > > We would also like to point out that in their original review, Reviewer 1 appears to take our work out of context, insisting on performance on dense graphs while also being overly dismissive of our results. For example, they took issue with an individual benchmark in which our method is not the best (DD), while simultaneously failing to consider all benchmarks in which our method outperforms. They previously referred to our performance on DD as “much worse” than the baseline DiffPool (81.36 vs. 85.88), while at the same time suggesting that LaPool “does not outperform baseline GIN that much” on the interpretability score (90.6 vs. 87.6).
> > >
> > > Since it appears that nothing we say will be evaluated objectively by Reviewer 1, we invite all other reviewers to compare results and explanations, and draw their own conclusions about the integrity of the work.

---

### Official Review · AnonReviewer3 · 2019-10-23
**Official Blind Review #3**

**Rating:** 6

**Review:**

The paper introduces a new pooling approach "Laplacian pooling" for graph neural networks, which the authors claim is able to better preserve information about the local structure, and to provide interpretability.  Namely, the pooling approach is based on finding centroids (nodes having high signal variation compared to their neighbors, via graph-Laplacian) and assigning other nodes to be "followers" based on a soft-attention mechanism. The authors add these new pooling layers to existing GNN architectures and show improved performance on problems of classification and generative modeling of molecular graphs. The paper also extends CNN interpretability techniques (integrated gradients) to GNNs.

I am borderline on the paper -- I'll give a weak accept rating for now. The proposed Laplacian-pooling ideas could be interesting, and the results encouraging -- but I found the mathematical motivation to be not very convincing, and has various (mostly correctable issues). The paper can be viewed as more of an engineering effort, which attempts to find practical tricks aimed at modeling molecular structures. What I like about the paper is that the authors make an earnest attempt to model the domain (biochemistry) -- for example they realize that a graph formalism for molecular structures is rather simplistic, and misses many important details -- such as different types of bonds (which require different types of edges). The authors also realize that typical neighborhood smoothing (diffusion) that makes sense for say spatial or social network graphs may not make sense for molecular graphs, where specific substructures (e.g. presence of a benzene ring) may be highly indicative for some classification tasks. The paper also contains a collection of interesting practical heuristics and observations (mainly in appendix) to help train GNN models for classification and generative modeling -- which researchers in the field may find valuable. I am not sure if the idea of coarsening (via hierarchical pooling) can be meaningfully applied to a wide-variety of natural molecules - but it does seem to make sense for some organic molecules -- e.g. protein chains.

Detailed comments:
1. The paper makes an analogy between band-pass filtering and the proposed approach. In my opinion the analogy is rather weak -- while it may carry over to spatial graphs (e.g. grid graphs) but may not apply to more complex graphs -- e.g. graphs where each node is at most a few hops away from any other node. It's not clear in what sense (3) corresponds to high-pass filtering. Can you show that it's somehow related to filtering-out the large (low-pass) eigenvalues of the graph-laplacian?

2. There is a typo in equation (1) -- equality of the quadratic form f'L f requires an unnormalized definition of the graph laplacian D - A, instead of I - D^{-1} A.

3. Notation in equation (3) -- is unclear and undefined.  What is ||L X||_{R^d} ? is that a norm (giving a scalar), or concatenation?  What is Top_k(V | L*S) -- the readers have to guess. You may be relying on notation from existing papers -- but still need to set up notation to be self-consistent. Subtle comment: "signal intensity variation" sounds like the variation of signal intensity -- e.g. something like difference of signal norms. Perhaps intensity of signal variation is a better term.  Sparsemax is also undefined. What is A^h -- the h-hop adjacency matrix?

4. I do not understand what do you mean by "information preservation" after pooling -- and I did not understand the importance of the "structure-aware feature content" definition, and the value of theorem 1.  Matrix C in the derivation is undefined.


**Experience Assessment:**

I have read many papers in this area.

**Review Assessment: Checking Correctness Of Derivations And Theory:**

N/A

**Review Assessment: Checking Correctness Of Experiments:**

I assessed the sensibility of the experiments.

**Review Assessment: Thoroughness In Paper Reading:**

I read the paper thoroughly.

---

> ### Author Response · Authors · 2019-11-09
> **Improved submission following reviewer's  suggestions.**
>
> We would like to thank the reviewer for the assessment of the paper and the helpful comments for improving the manuscript. We have taken the issues raised into consideration and have modified the manuscript accordingly.
>
> While the idea behind Laplacian pooling certainly arose out of a desire to better model molecular structures, we do believe it represents a useful contribution and tool for graph representation learning in general, particularly in cases where graph density decreases and where other approaches may not be well suited. Even in the case of social network graphs, where neighborhood smoothing is less of an issue, one could imagine that delineating between user clusters of the graph based on the signal variation may bias the learning process in a helpful manner.
>
> We address your numbered comments below:
>
> 1.  We agree with the reviewer regarding the analogy between our approach and GSP and have now clarified in the manuscript by adjusting these claims.  It does however guide the intuition behind the choice of the centroid selection strategy.  Indeed, Eq. 3 selects a subgraph,  among the set of possible subgraphs (not necessarily induced) of the same size on the graph G, containing nodes where the signal varies the most. This ensures that we are retaining the highest amount of total variation on the graph, under the constraint of node selection.
> An alternative view would be to consider the effect of that choice in the Fourier domain. For simplicity, let’s assume that we have a 1-D signal on the graph. The Laplacian can be written as UλU^T, where U are the eigen vectors and λ the eigen values.  A filter h on the graph can generally be seen as a function (often polynomial) which modifies the frequency coefficients by acting on the eigenvalues: H= h(L) = Uh(λ)U^T.  This filter is then applied to the signal X  as HX. The general step can be described as first translating the signal into the spectral domain, modifying it using the frequency (eigenvalues), then bringing it back to the spatial domain. In our case, we are interested in the part of the graph where the signal varies the most. This can be seen as applying a filter which gives more weights to the highest frequencies (high eigenvalues). See https://arxiv.org/pdf/1307.0468.pdf and http://citeseerx.ist.psu.edu/viewdoc/download?doi=10.1.1.367.6064&rep=rep1&type=pdf for additional details.
> Although this roughly explains the intuition, as the reviewer points out, it does not indicate that we are indeed performing a high pass filtering, given the constraint of retaining graph nodes, and the additional dynamics of downsampling the graph before and after the pooling layer.
>
> 2. This is correct, we have corrected the typo in the manuscript.
>
> 3. We agree that the notation is unclear, and have added detail and explanation (in particular for Equation 3).
>
> 4. This section has generated confusion and its value was indeed overstated. We have elected to remove it from the manuscript.

---

### Official Review · AnonReviewer2 · 2019-10-23
**Official Blind Review #2**

**Rating:** 6

**Review:**

Summary

The authors propose a new pooling layer, LaPool, for hierarchical graph representation learning (Ying et al., 2019) by clustering nodes around centroids that are selected based on "signal intensity variation". The signal intensity variation of node x is defined as sum_{y in HOP(x, h)} ||x - y||  where HOP(x, h) is the set of nodes reachable from x within h hops. Once top k maximizers are selected as centroids (k can be predetermined or dynamically chosen), a sparse cluster assignment distribution is computed for each node using sparsemax (Laha et al., 2018), and the affinity matrix and the node embeddings are coarsened as in Ying et al. (2019). The authors show that LaPool can improve performance in various graph-related tasks over baselines and generate interpretable clusters.


Strengths

- Explicit centroid selection based on signal intensity variation seems like an intuitive idea worth investigating.

- LaPool seems to be empirically effective, in particular outperforming Graph U-Net which is probably the most relevant baseline (also k-max pooling for hierarchical graph representation learning). But I'm not an expert on the considered tasks, so I cannot judge how significant these results are.


Weaknesses:

- The paper has issues with clarity. There seem to be many sloppy notations as well as unclear (possibly wrong) arguments.

1. Isn't equation (1) true for unnormalized graph Laplacian (D - A), not normalized?

2. In the proof of Theorem 1, why is it that C C' X = X (i.e., X is in range(C))? This is a crucial step that I'm not sure why is true. I might be missing something, but it's not clear to me in the current version.

3. Sloppy notations. What do "A^h", "||L X||_{R^d}", "top_k(V|L S)" mean in equation (4)? I can infer their meaning, but do I have to? This also interferes with my other confusions: back to question #1, is it true that S = ||L X||_{R^d}?

4. Is the number of clusters fixed or dynamic for these experiments? Figure 2 seems to be dynamic based on the main text, but I cannot tell if it's the case for other experiments. Based on the provided code, it seems the default number of clusters is always 10% of the number of nodes.

- The paper doesn't do a good job of contextualizing itself. Graph U-Net seems to be the most relevant previous work, but there's no discussion of how this work relates and why it's better. For instance, is it the case that Graph U-Net is not interpretable because there's no explicit clustering? It'd be much clearer to spell out such differences.

- It's unclear why certain design choices are made and why they're better. For instance, is swapping entropy minimization with sparsemax necessarily better? I understand not every design decision should be (or needs to be) justified, but it's helpful if it is to understand what helps and what doesn't.


Summary

The core idea of the paper, clustering nodes based on signal intensity for hierarchical graph representation learning, is interesting and seems to be useful in practice, but the paper has issues with clarity and the rest of the framework is a bit limited in novelty (DiffPool + sparsemax + GIN).

**Experience Assessment:**

I have read many papers in this area.

**Review Assessment: Checking Correctness Of Derivations And Theory:**

I assessed the sensibility of the derivations and theory.

**Review Assessment: Checking Correctness Of Experiments:**

I assessed the sensibility of the experiments.

**Review Assessment: Thoroughness In Paper Reading:**

I read the paper at least twice and used my best judgement in assessing the paper.

---

> ### Author Response · Authors · 2019-11-09
> **We have clarified the notation and have added new results for Graph U-net**
>
> We thank the reviewer for their assessment of our manuscript and have revised it accordingly to address the points raised, in particular regarding clarity and notation.
>
> We agree that we may not have been clear in positioning Lapool in the field and relating it to previous work. We have updated Section 3.4 along with the discussion and the conclusion of the main text to better highlight the differentiation achieved by LaPool, and hope that in doing so we have clarified the significance of this work in relation to the existing literature.
>
> In short, we believe pooling methods are powerful tools able to increase the effectiveness of  GCNs. In contrast with previous methods, LaPool focuses its pooling to encourage the learning algorithm to find important substructures, as determined by the Laplacian, and clusters nodes together based on these substructures. This method thus achieves the goals of:
>
> 1) Preserving connectedness between underlying structures (as opposed to Graph U-net and DiffPool).
> 2) Coarsening the graph based on existing centroids to identify substructures (as opposed to Diffpool).
> 3) Preserving node feature information after coarsening (as opposed to Graph U-net).
> 4) Considering structural constraints of the molecular graphs (as opposed to both Diffpool and Graph U-net).
>
> Furthermore, following the reviewer’s comment on Graph U-Net, we have included interpretability experiments with this method. Interpretability experiments were previously limited to the baselines that were also performing graph downsampling via clustering, namely DiffPool, to also compare the quality of cluster assignment.
>
> We have previously observed that the graph decimation performed by Graph U-Net often disconnects molecular graphs, generating connected components (or isolated nodes) that can no longer pass information, which is critical for capturing larger subgraphs. We now show this fact on Figure 2.  The intuition for this is based on the fact that the node selection scheme of Graph U-Net solely uses node features. While this approach holds on interaction networks due to nodes with similar features often being neighbors, on molecular graphs such a hypothesis fails, resulting in nodes being selected across the full graph. Moreover, since edges are not considered, the downsampled graph becomes disconnected.
>
> Our detailed answer follows.
>
> 1. The reviewer is correct. This is a typo and we have updated the manuscript accordingly.
>
> 2. See Point 4 of Reviewer #3. The entire section has been removed due to lack of clarity and confusion.
>
> 3. We have improved the notation in the revised manuscript and explained each of these terms.
>
> 4. Results reported for LaPool include both dynamic and fixed clustering. Cluster size selection was considered as a hyperparameter (same for DiffPool and Graph U-Net), while experiments for Figure 2 were conducted solely with the dynamic clustering method. This is now clarified in the main text. In the source code provided, the dynamic selection is used when the hidden_dim is not defined (-k can be overridden by --hparams for parameters specific to hierarchical pooling layer)
>
> Finally, on the reviewer’s comment regarding the use of a sparsemax instead of the entropy minimization:
>
> As noted by the DiffPool authors, training with the side loss is useful for making the assignment matrix more sparse and thus helps to improve interpretability. We found that this does not necessarily result in better performance, and also takes longer to converge. For example, on molecular graphs shown on Figure 2, despite the entropy regularization, the clustering performed by DiffPool remains extremely fuzzy. We achieved better results with the sparsemax operator, while removing the need for the entropy regularization to attain interpretability.

---

### Decision · Program_Chairs · 2019-12-19

**Decision:**

Reject

**Comment:**

The paper introduces a new pooling approach "Laplacian pooling" for graph neural networks and applies this to molecular graphs. While the paper has been substantially improved from its original form, there are still various concerns regarding performance and interpretability that remain unanswered. In its current form the paper is not ready for acceptance to ICLR-2020.